# Phenotyping Anther Extrusion of Wheat Using Image Analysis

**Zachary James Winn** [1,2,*], **Dylan Lee Larkin** [2,3], **Jamison Trey Murry** [2,4], **David Earl Moon** [2]
and **Richard Esten Mason** [2,5]

1 Crop and Soil Sciences, North Carolina State University, Raleigh, NC 27607, USA
2 Crop, Soil, and Environmental Sciences, University of Arkansas, Fayetteville, AR 72701, USA;
dylan.larkin@aardevo.com (D.L.L.); Jamison.Murry@usda.gov (J.T.M.); demoon@uark.edu (D.E.M.);
Esten.Mason@colostate.edu (R.E.M.)
3 Aardevo North America, Boise, ID 83706, USA
4 USDA Natural Resource Conservation Service, Pine Bluff, AR 71601, USA
5 Soil and Crop Sciences, Colorado State University, Fort Collins, CO 80523, USA
* Correspondence: zjwinn@ncsu.edu

**Abstract:** Phenotyping wheat (*Triticum aestivum* L.) is time-consuming and new methods are necessary to decrease labor. To develop a heterotic pool of male wheat lines for hybrid breeding, there must be an efficient way to measure both anther extrusion and the size of anthers. Five hundred and ninety-four soft red winter wheat lines in two replications of randomized complete block design were phenotyped for anther extrusion, a key trait for hybrid wheat production. A device was constructed to capture images using a mobile device. Four heads were sampled per line when anthesis was evident for half the heads in the plot. The extruded anthers were scraped onto a surface, their image was captured, and the area of the anthers was taken via ImageJ. The number of anthers extruded was estimated by counting the number of anthers per image and dividing by the number of heads sampled. The area per anther was taken by dividing the area of anthers per spike by the number of anthers per spike. A significant correlation ($R = 0.9$, $p < 0.0001$) was observed between the area of anthers per spike and the number of anthers per spike. This method is proposed as a substitute for field ratings of anther extrusion and to quantitatively measure the size of anthers.

**Keywords:** image analysis; anther extrusion; imaging methods

## 1. Introduction

Without substantial change in the developed world's diet, current yield trends for wheat (*Triticum aestivum* L.) are not on track to meet the projected caloric demands of 2050 [1,2]. Global grain yield must increase by approximately one petagram from the 2007 production benchmark if we are to meet the projected demands. However, the gain per year has slowed to a near stagnant pace [1]. A proposed method for overcoming this deficit is to take advantage of heterosis and convert wheat varieties from an inbred line breeding system to an $F_1$ hybrid breeding system [3–5].

There are several steps to developing a successful hybrid wheat cultivar. First, breeders must separate lines into genetically distinct heterotic pools grouped by gender and genetic distance [6]. Female wheat parents should be shorter than males, receptive to pollen, and flower later. Male wheat parents should be taller than females, exhibit high anther extrusion, produce large pollen-baring anthers, and flower earlier. To develop these heterotic pools, breeders must select for these traits within these heterotic pools via reciprocal recurrent selection [6,7].

Currently, a limiting factor in hybrid wheat production is the lack of male parents exhibiting high anther extrusion. This is assumed to be a byproduct of selection for closed flowering traits [4,7]. Measuring male anthesis traits can be laborious and time-consuming. Although anther count measurements can be obtained within the field, this can only be done on a limited number of accessions within the timeframe of anthesis [3,4].

In the present study, 594 soft red winter wheat lines from the southeastern United States were phenotyped via image analysis for the area of anthers extruded. We propose a new method of analyzing the number of anthers extruded per spike (NOAPS) by using the area of anthers extruded per spike (AOAPS) detected through image analysis as a bi-proxy index. Through this medium-throughput analysis technique, we hope to remove biases from visual ratings and find a new selection criterion for male hybrid parents.

## 2. Materials and Methods

### 2.1. Plant Materials

Portions of the materials and methods for this study were previously described by Winn (2019) [8]. The Historic Gulf Atlantic Wheat Nursery (HGAWN) was planted in two replications using a randomized complete block design in Fayetteville, Arkansas. The HGAWN population was originally derived by Sarinelli et al. (2019) for genomic selection and it consists of 594 RILs; of these RILs: 103 were produced by the University of Arkansas, 105 from the University of Georgia, 109 from Louisiana State University, 104 from North Carolina State University, 60 from Texas A&M University, 41 from University of Florida, 44 from Virginia Institute of Technology, 19 from Clemson University, and 9 from private industry or the United States Department of Agriculture-Agricultural Research Service (USDA-ARS) [8,9].

Genotypes were planted in 1.2 m long single-row plots with 40 cm spacing between neighboring plots. Soil sampling was performed and an application of 67 kg Ha$^{-1}$ of urea was done in late February, followed by another application of nitrogen at 33 kg Ha$^{-1}$ in late March. Harmony® Extra was applied at 0.28 kg Ha$^{-1}$ in early February to control various broadleaf weeds and Axial® was applied in early March at a rate of 0.6 kg Ha$^{-1}$ to control for annual ryegrass (*Lolium multiflorum* L.) weeds.

### 2.2. Phenotyping

Heading date (HD) data was collected when 50% of the plot was observed to be heading from mid-April until all plots had headed out. Anthesis date (AD) was taken on the day that anthesis was evident in half of the plot. Both HD and AD were recorded in days from January 1st. Heads used for anther imaging were sampled at AD.

To obtain the AOAPS four heads were collected per genotype at AD. Collected heads were placed into glassine envelopes and stored at 4° C; all samples were imaged no later than 48 h after sampling. Extruded anthers were gently swept off heads by hand onto a red surface and then imaged. NOAPS was estimated by counting the number of anthers in each image and dividing by the total number of spikes available. The area per anther (APA) was calculated by taking the total AOAPS and dividing by the NOAPS [8].

### 2.3. Imaging

An image capturing device was constructed for the imaging of anthers. The goal of this imaging device was to capture images at a specified height with consistent resolution and lighting at a minimal cost (Figure 1).

To construct the mobile imaging device, a 4.5 cm wide hole was drilled in the center of a 4.7 L Little Giant® feed pan (Miller Manufacturing Company; Eagan, MN, USA) to accommodate the lens of the camera. RUST-OLEUM® Camouflage Ultra Flat Black (Rust-Oleum USA; Vernon Hills, IL, USA) spray paint was applied to the interior of the feed pan to create a non-reflective matte surface. A handle was attached just above the camera opening to prevent the mobile device from changing position during imaging. The hook portion of a 15.25 cm length of hook-and-loop fastener was attached approximately 6 cm from the 4.5 cm opening to prevent the movement of the mobile device during imaging and provide a consistent shot.

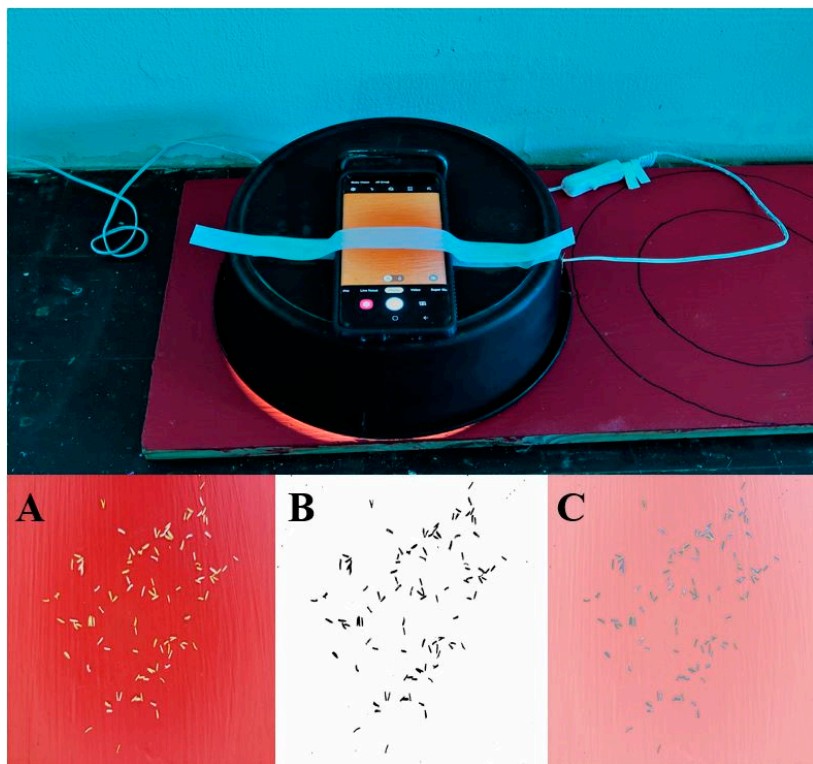

**Figure 1.** The imaging device used for image capture (**A**) original image, (**B**) area of detected anthers, and (**C**) an overlay of the two images.

A section of tape lights from a GoodEarth® Self-Adhesive Tape Lighting Kit (Good Earth Lighting; Mount Prospect, IL, USA) measuring approximately 63.5 cm was fixed to the interior of the pan. An imaging stage measuring 132 cm by 23 cm was painted with Classic Red Valspar Ultra® Interior Flat Paint (Valspar; Cleveland, OH, USA) until the paint was completely opaque and the texture of the board was no longer visible. Images were taken on a Samsung® S9+ (Samsung; Seoul, South Korea) phone with the following camera specifications: a 12-mega pixel camera, with a focal length of 4 mm, a F-stop of f/1.5, a shutter speed of 1/125 s, at an ISO of 50, and a white balance between 2000–3000 Kelvin [8].

### 2.4. Image Analysis

Images were analyzed via java script in ImageJ software version 1.5 [10]. A macro was composed for image analysis. The macro contained the following procedures: setting a scale of pixels to a known distance, adjusting the brightness and contrast, applying a gaussian blur, selecting a specific color threshold, and analyzing the selected particles in the image.

A reference image with a ruler was taken, and the ImageJ line function was then used to set 320 pixels to 10 mm$^2$. The brightness and contrast minimum and maximum were set to 130 and 255, respectively. A gaussian blur of $\sigma = 2$ was applied. A color threshold was then selected using the hue, saturation, and brightness settings.

Hue was set to a minimum of 1 and a maximum of 255, saturation was set to a minimum of 0 and a maximum of 255, and brightness was set to a minimum of 142 and a maximum of 255. Selected particles were then analyzed and holes between selected pixels were included. The area of selected pixels was reported in mm$^2$ [8].

### 2.5. Statistical Models and Software

All models were done in R statistical software version 4.0.3 using the package "asreml" version 4 [11,12]. All graphics and visuals were generated using the packages "ggplot2",

"plotly", and "GGally" [13,14]. A series of mixed linear models (MLM) were employed to analyze HD, AD, AOAPS, NOAPS, and APA. The formula used is as follows:

$$y_{ij} = \mu + G_i + R_j + \varepsilon_{ij}$$

where $y$ is the response variable, $\mu$ is the population mean, $G$ is the fixed genotype effect, $R$ is the random effect of replication $\left( R \sim N(0, I\sigma_R^2) \right)$, and $\varepsilon$ is the residual error ($\varepsilon \sim N(0, I\sigma_\varepsilon^2)$). Repeatability was estimated by dividing the variance of the genotype effect by the total phenotypic variance. The best linear unbiased estimates (BLUEs) of the fixed genotype effect were obtained through the function "asreml.predict()" in the package "asreml" for later use in correlation and regression studies. A principal component analysis (PCA) was performed on the eigen value decomposed correlation matrix derived from the BLUEs using the base function "prcomp()" from the "Stats" package in R in order to assess if there was any discernable pattern of clustering between variables.

## 3. Results

Of the total possible observations planted, 835 were collected and BLUEs for all traits were obtained for 491 of the total 593 RIL lines of the HGAWN. All response variables in the MLM were significant for the genotype effect ($p < 0.05$) and estimates of the genotype variance for each trait were obtained to estimate repeatability and the standard error of those repeatability estimates (Table 1).

**Table 1.** Listed summary statistics derived from mixed linear models. *p*-values are derived from a chi-squared distribution. A *p*-value of 0.05 or less is considered significant.

| Trait | Degrees of Freedom | Sum of Squares | Wald Statistic | *p*-Value | *R* * | Standard Error ** |
|---|---|---|---|---|---|---|
| Heading Date | 580 | 1761.10 | 1194.58 | <0.0001 | 0.28 | 0.08 |
| Anthesis Date | 492 | 928.24 | 777.87 | <0.0001 | 0.24 | 0.06 |
| Area of Anthers per Spike | 494 | 15,322.21 | 636.92 | <0.0001 | 0.17 | 0.27 |
| Number of Anthers per Spike | 492 | 58,067.08 | 607.61 | 0.0003 | 0.17 | 0.89 |
| Area per Anther | 491 | 7.47 | 1299.32 | <0.0001 | 0.49 | 0.04 |

* = Repeatability. ** = Standard error in reference to repeatability.

The best linear unbiased estimates obtained from the fixed mixed linear models were used to perform a correlational analysis (Figure 2). A highly significant correlation ($R = 0.9$, $p < 0.0001$) was found between the AOAPS and NOAPS. No significant correlation was found between the NOAPS and APA. However, a modest positive correlation ($R = 0.47$, $p < 0.0001$) was found between AOAPS and APA. Significant correlation coefficients were seen between HD, AD, AOAPS, NOAPS, and APA. To assess the relationship between the phenotypic measurements obtained, a PCA was conducted on the correlation matrix of BLUEs to observe any discernable pattern. The first, second and third principal components accounted for 46.5%, 32.4%, and 16.1% of the total variation, respectively. When the first three principal components were plotted, HD and AD grouped together, while NOAPS and AOAPS grouped together, and APA was alone, which may indicate that the AD and HD do not interact with NOAPS and AOAPS (Figure 3).

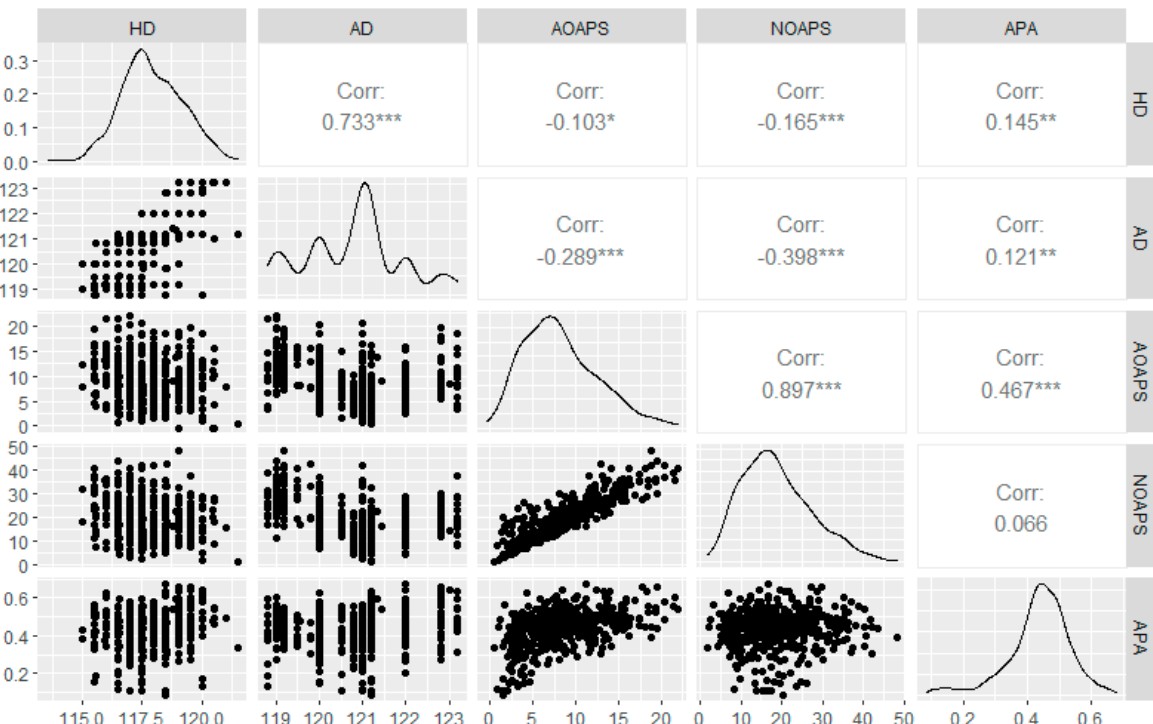

**Figure 2.** A pairs plot of the BLUES derived from the linear models. The lower triangle displays the scatterplots of each comparison with the population mean labeled with a red dot. The diagonal displays the histogram of the labeled trait with an overlayed density curve. The upper triangle displays the Pearson's correlation coefficient. Any coefficient without an asterisk is insignificant ($p > 0.05$). Otherwise, asterisks denote the following: $* = p \leq 0.05$, $** = p \leq 0.01$, $*** = p \leq 0.001$.

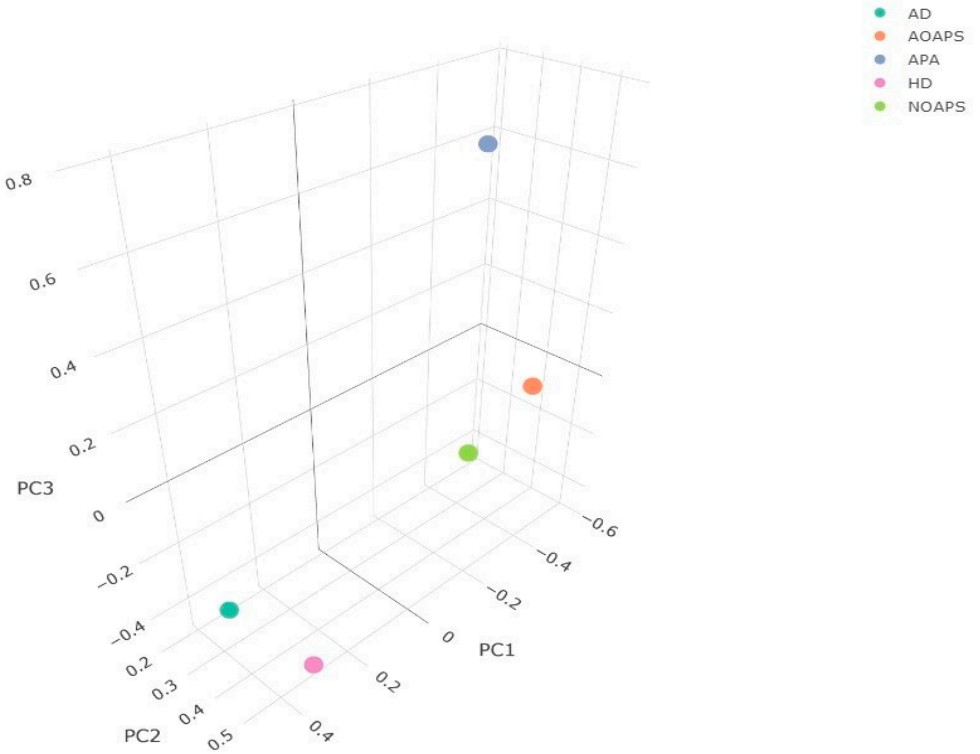

**Figure 3.** Three-dimensional principal component scatterplot. Each color dot represents a trait analyzed as denoted in the figure legend in the top-right corner. Each axis is labeled with the principal component they represent.

## 4. Discussion

Hybrid breeding may be a possible method for increasing wheat yields, and further study is required. The production of male wheat breeding lines exhibiting high levels of anther extrusion is a crucial step that must be taken to begin effective hybrid wheat breeding. While anther extrusion in biparental populations is estimated to have a high heritability [15], obtaining ratings for anther extrusion can be time consuming and visual rating is confounded by human bias. The present study's objective was to provide an alternative phenotyping method that lowered the influence of bias for the evaluation of male anther extrusion in wheat.

In this study, the NOAPS, AOAPS, and APA were significant relative to the genotype, and the NOAPS and AOAPS showed a strong relationship. This suggests that this method may be sufficient to serve as a substitute for physically counting anthers. The repeatability of NOAPS and AOAPS were highly similar, yet the standard error of the repeatability for NOAPS was far higher than AOAPS. This indicates that while both methods have a large standard error in terms of repeatability due to the limited replications in this study, AOAPS appears to have a significantly lower standard error, implying that its repeatability over many site-years may be higher than NOAPS.

In reference to the high standard error of AOAPS's repeatability: the sampling that was applied to image the extruded anthers onto the sampling area may have deformed the anthers due to the sweeping of anthers off the wheat head. Perhaps the AOAPS's repeatability could be improved using a gentler method of anther removal.

The APA was significant for the genotype effect and had both the highest repeatability and the lowest standard error for repeatability. The area of anthers has had limited study. However, the broad-sense heritability of anther size is estimated to be relatively high [16]. Regardless, the methods used to measure this trait are highly laborious and time consuming.

Furthermore, we did not detect a significant correlation between the APA and the NOAPS. This may indicate that in this population, the number of anthers and size of anthers are not correlated. Wheat anthers size has been positively correlated to the number of pollen granules in wheat, and the length of wheat anthers has been shown to be variable in separate populations [17]. However, there is a lack of studies relating the size and number of anthers on singular heads.

We did not have a direct comparison in this study to detect if the APA is highly correlated with the anther size. However, this represents a route of future study for this type of image analysis. Perhaps the use of machine learning to determine the number of anthers in the image could reduce the time to count anthers and produce an approximation of the APA without the need for physically counting the number of anthers in each picture. Further replications, applications, and studies are therefore required to validate this system for the use of measuring APA.

## 5. Conclusions

Phenotyping wheat is time consuming and new methods are needed. Anther extrusion is an important trait for male line development in hybrid wheat breeding. In the current study, we have demonstrated a new method of measuring the number of anthers extruded, which can quantify the size of anthers and may act as a possible substitute for the physical counting of anthers.

**Author Contributions:** Conceptualization, Z.J.W.; methodology, Z.J.W.; software, Z.J.W.; validation, Z.J.W.; formal analysis, Z.J.W.; investigation, Z.J.W., J.T.M., and D.L.L.; resources, Z.J.W. and R.E.M.; data curation, Z.J.W., D.L.L., and J.T.M.; writing–original draft preparation, Z.J.W.; writing–review and editing, Z.J.W., R.E.M., D.L.L., J.T.M., and D.E.M.; visualization, Z.J.W.; supervision, R.E.M. and D.E.M.; project administration, R.E.M.; funding acquisition, R.E.M. All authors have read and agreed to the published version of the manuscript.

**Funding:** This research was funded by USDA National Institute of Food and Agriculture, grant number 2017-67007-25939.

**Institutional Review Board Statement:** Not applicable.

**Informed Consent Statement:** Not applicable.

**Data Availability Statement:** Data is available upon request via contact with the corresponding author and at <https://github.com/zjwinn/Phenotyping-Anther-Extrusion-of-Wheat-Using-Image-Analysis>.

**Acknowledgments:** This work is supported by the Agriculture and Food Research Initiative Competitive Grant 2017-67007-25939 (Wheat-CAP) from the USDA National Institute of Food and Agriculture.

**Conflicts of Interest:** The author claims no conflict of interest.

## Abbreviations

| | |
|---|---|
| NOAPS | Number of Anthers Extruded Per Spike |
| AOAPS | Area of Anthers Extruded Per Spike |
| APA | Area per Anther |
| HD | Heading Date |
| AD | Anthesis Date |
| HGAWN | Historic Gulf Atlantic Nursery |
| BLUEs | Best Linear Unbiased Estimates |
| PCA | Principal Component Analysis |

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
