# Peer review of "Phenotyping Anther Extrusion of Wheat Using Image Analysis"

_agronomy, doi:10.3390/agronomy11061244_

Round 1
Reviewer 1 Report
Dear Authors of "Phenotyping Anther Extrusion of Wheat using Image Analysis",
I have with great interest read your manuscript and i have the following comments/suggestions:
Line 135 - "y is the observation"+ why is y not the resonse variable?
I can't find any information regarding the model to predict area? the method is described, but no results? it could be variation between replicate images of same sample or bias.
Why did you use linear models when most scatterplots in figure 2 clearly shows that the relationsships are not linear?
X- and Y-axis titles are missing in figure 2
Line 143-144 - the heading should be numbered, use only normale not capital letters and moved to the left
Figure 3 - Is it from one sample? or why is there only one value for AD, AOAPS, APA, HD and NOAPS?
A conclusion is missing, it is in the abstract but 5. Conclusions is empty
Author Response
Hi, Thanks for reviewing my paper in such a timely manner. I have attached my response to your concerns in the word document.

Reviewer 2 Report
The authors have done a good job of presenting the results and methodology. Just curious about not having a correlation between NOAPS and APA.
Delete "for" in line 72.
Author Response
I am wondering why there is a lack of a correlation there as well. I can speculate on the possible reasons for this in the discussion. I believe that this might just indicate that there is not a correlation between the size of anthers and the number of anthers in this population. I almost expected to see a negative correlation there (more anthers, less area per anther) but that is not evident. I will also delete for in line 72 prior to resubmission.
Reviewer 3 Report
The manuscript showed phenotyping method for anther extrusion in wheat. Topic is interesting and fit for the aims and scope for the journal. However, the manuscript contains a serious weakness. My main concern is reliability of the method. The authors didn’t show any verification of the method. Although they conducted MLM and PCA, the data isn’t meaningful for showing reliability of the method. They should compare their data with the data from other methods. Alternatively, they can conduct QTL analysis for anther extrusion to check whether known QTL can detect from their data.
Author Response
I think that by counting the number of anthers per plot and regressing that number to the area derived from image analysis, I have shown how well correlated those two traits are. I think by counting the number of anthers per plot and analyzing the area and correlating them, I have shown that the area of anthers and the number of anthers share a strong linear relationship. I think that this implies that this method could be used in lieu of manually counting the anthers. This paper is to demonstrate the method to get the area of anthers.
I have published a thesis (An Evaluation of Hybrid Traits, Yield, and Major QTL Effect on Heterosis in Hybrid Soft Red Winter Wheat (uark.edu)) where I use genome-wide SNP markers to conduct a GWAS for area of anthers per spike and number of anthers per spike in this same population, and I found that they both pick up known MTA for anther extrusion . However, I chose to not include that information in this publication due to the lack of replication required for a GWAS.